# Host Microbiomes Influence the Effects of Diet on Inflammation and Cancer

**DOI:** 10.3390/cancers15020521

**Published:** 2023-01-14

**Authors:** Ramsha Mahmood, Athalia Voisin, Hana Olof, Reihane Khorasaniha, Samuel A. Lawal, Heather K. Armstrong

**Affiliations:** 1Department of Internal Medicine, University of Manitoba, Winnipeg, MB R3E 3P4, Canada; 2Department of Medical Microbiology and Infectious Diseases, University of Manitoba, Winnipeg, MB R3E 3P4, Canada; 3Department of Immunology, University of Manitoba, Winnipeg, MB R3E 3P4, Canada; 4Department of Food and Human Nutritional Sciences, University of Manitoba, Winnipeg, MB R3E 3P4, Canada

**Keywords:** microbiome, diet, dysbiosis, tumor microenvironment, inflammation, cancer

## Abstract

**Simple Summary:**

This review provides an update on recent evidence supporting the role of various microbiomes mediating the interactions that occur between dietary factors, inflammation, and various cancers. Microbiomes interact with localized and systemic host cell populations where they help to maintain immune homeostasis. Microbiota use different dietary factors for energy and in turn produce by-products that impact the host cell populations. Dietary factors can also influence the composition and diversity of microbiota populations, in turn impacting the interactions of the microbiomes with host. Perturbations in this system, commonly referred to as dysbiosis, have been associated with various diseases including cancer.

**Abstract:**

Cancer is the second leading cause of death globally, and there is a growing appreciation for the complex involvement of diet, microbiomes, and inflammatory processes culminating in tumorigenesis. Although research has significantly improved our understanding of the various factors involved in different cancers, the underlying mechanisms through which these factors influence tumor cells and their microenvironment remain to be completely understood. In particular, interactions between the different microbiomes, specific dietary factors, and host cells mediate both local and systemic immune responses, thereby influencing inflammation and tumorigenesis. Developing an improved understanding of how different microbiomes, beyond just the colonic microbiome, can interact with dietary factors to influence inflammatory processes and tumorigenesis will support our ability to better understand the potential for microbe-altering and dietary interventions for these patients in future.

## 1. Introduction

Sustained inflammation is a hallmark of cancer, predisposing the host to tumorigenesis and promoting all stages of tumor progression [1,2,3]. Significant advancements have been made in recent years as research has continued to uncover the role of various host cell subsets (e.g., leukocytes, cancer stem cells, innate lymphoid cells), environmental factors, and microbiomes in development and progression of cancers [1,3,4,5,6,7]. Interestingly, one environmental factor that continues to receive mounting, yet often conflicting evidence for its role in cancer, is diet [8,9]. Many studies offer simplified approaches, for example, suggesting that a “Western” diet increases risk of malignancies, while a “Mediterranean” diet is thought to reduce inflammatory burden and tumorigenesis [10,11,12,13]. Progress beyond correlative analysis has demonstrated that the high phenolic contents of extra virgin olive oil, a major component of the Mediterranean diet, confers protection in an inflammatory setting by reducing the effects of pro-inflammatory mediators partially through interactions with the gut microbiota [14,15,16]. This is just one example of how mechanistic research continues to underpin the complex role that microbiomes play in mediating host interactions with dietary factors and how we can harness this knowledge to better understand the processes of inflammation and tumorigenesis in several organ systems.

## 2. The Impact of Diet on Cancer

In recent decades, the impact of diet on cancer has attracted considerable attention related to either cancer prevention or adjuvant therapy. Dietary intake is thought to contribute to 30–35% of cancer incidence; however, the influence of dietary factors in specific types of cancer such as colorectal cancer can be even higher [8,17,18]. One of the most impactful recent studies of the implications of dietary factors in various cancers examined associations between diet and risk of cancer at 11 anatomical sites, evaluating the inherent biases present in many studies and the downfalls associated with present methods used to assess diet [9]. They found several concrete pieces of evidence relating dietary factors to cancer outcomes based on studies published to date. Alcohol consumption remained one of the most positively associated risk factors for a variety of cancers including esophageal, head and neck, colorectal, liver, and post-menopausal breast cancer [9]. Consumption of coffee was inversely linked to risk of liver cancer and skin basal cell carcinoma [9]. Lastly, consumption of calcium, whole grains, and dairy products was inversely linked to risk of colorectal cancer [9].

However, the study of diet is complex, as a majority of studies evaluate real-world diets which involve a high degree of heterogeneity. The key components included in Mediterranean, Western, Paleolithic, and ketogenic diets and their impacts on gut microbiota have been described previously which Sinibaldi et al. highlighted in Figure 1 of their manuscript [19]. Whole food diets such as the Mediterranean diet (MD) consist of a recommended intake of dietary carbohydrates (including fibers; ~30 g/day), high intake of mono-unsaturated fatty acids, polyphenols, and omega-3 fatty acids which have been shown to reduce inflammatory signaling and elicit protective effects against cancer [20,21,22]. In contrast, the ketogenic diet (KD), which consists of very low carbohydrate intake (including fibers), high fat intake, and adequate protein intake to limit glucose levels, has long been recognized for its benefits in neurological conditions where KD reduces seizure frequency [23,24]. However, there remains inadequate scientific evidence to support long-term safety of KD. While KD has more recently received attention for a variety of health conditions, with some evidence to suggest it may elicit anti-tumor effects in certain cancers, the improvements are often only temporary, and this restrictive diet negatively impacts diet quality by recommending increased intake of foods linked to chronic disease and cancer risk while in turn decreasing intake of protective foods [23,24]. The Paleolithic diet (PD) displays similarities to the MD, as it is characterized by high consumption of fruits, vegetables, lean meats, fish, eggs, nuts, and seeds, yet it excludes all processed foods, legumes, grains, dairy products, and plant oils (except for olive and coconut oil) [25,26]. The PD also consists of extraordinarily high amounts of fiber intake (~100 g/day), yet interestingly, while the benefits of PD may be in part due to high consumption of microbiota-accessible carbohydrates which positively impact gut microbiota diversity, there is some evidence to suggest PD promotes higher microbiome diversity compared to MD [22,27].

While evidence suggests that diet plays a role in ~30% of all cancers, whether diet alone or in conjunction with other interventions can be used in the treatment cancers remains to be fully understood [8,17,18]. Many of the dietary factors increased in diets that are shown to improve cancer outcomes, including fruits, vegetables, dietary fibers, and proteins, partially rely on interactions with gut microbiota to elicit health benefits [12,13]. These microbiota communities, particularly oral and intestinal microbiota, are capable of utilizing dietary factors to produce by-products that impact host cells, and in turn dietary factors can also influence growth and health of microbiota [12,13,28]. Dietary metabolites such as short chain fatty acids (SCFAs) produced during fiber fermentation by select commensal microorganisms in the intestine play a protective role against cancers by inhibiting myeloid cell-driven pro-tumorigenic inflammation [29]. Furthermore, alterations in the Firmicutes/Bacteroidetes ratio which are particularly high in obese individuals and patients with colorectal cancer can be reduced by decreasing caloric intake [30,31]. Within the gut, microbiota can also mediate accessibility of polyphenols, a major component of fruits and vegetables which are metabolized into their active forms and have been known to play a role in cell cycle arrest and apoptosis, as well as the prevention of production of inflammatory cytokines [32,33]. Several vitamins including biotin, cobalamin, folate, and niacin amongst others have also been shown to elicit anti-tumor effects; several of these vitamins can be produced by microorganisms in nutrient-scarce conditions [34,35,36,37]. Select microbes also contribute to anti- or pro-inflammatory responses which can influence the development and progression of various cancers, demonstrating the importance of promoting health of these microbiota communities [12,13,28].

## 3. The Role of Microbiomes in Cancers in Connection to Diet

Several studies have alluded to the role of microbiomes in tumorigenesis and cancer immunotherapy [38,39,40]. Infectious agents including viruses, bacteria, and fungi are thought to cause over 20% of global cancer cases [6,41]. Commensal microorganisms regulate tumor suppression, engage in immunosurveillance during tumor development, and play a significant role in the maturation and stimulation of the immune system [12,13,42,43]. For example, members of the *Lactobacillaceae* family are involved in anti-tumor immune responses driven through stimulation of dendritic cell maturation and subsequent acquisition of other immune cell subsets, including anti-tumorigenic myeloid cells [42,43]. On the other hand, pathogenic microorganisms play a key role in tumor development and metastasis and impair responses to anti-cancer therapies [39]. For example, infectious agents including human papillomavirus (HPV), hepatitis C virus (HCV), hepatitis B virus (HBV), and *Helicobacter pylori* have significant and unrefuted roles in tumorigenesis [44].

However, the microbial communities that make up human microbiomes are more complex than the individual microbes that have been implicated in cancers; each microbiome includes a variety of commensal, pathobiont, and pathogenic microorganisms along with microenvironment factors including various metabolites [6]. Together these microbiome communities influence the host directly and indirectly, resulting in beneficial and detrimental effects on drug metabolism, hormone regulation, inflammation, nutrient access, and uptake [6,45]. The cancer ecosystem involves a dynamic interaction between the related microbiomes, cancer cells, non-cancerous host cells, circulating metabolites, and systemic immunity [6]. Given their diverse nature, microbiomes are in a constant state of flux and are influenced heavily by nutrition, amongst other external factors [46]. Different dietary factors have been implicated in various cancers, but how the host microbiomes impact or influence the effects of these dietary factors on cancer development and progression requires further elucidation. The gut microbiome has been and continues to be established as a significant regulator of health and disease; however, it is not the only microbiome involved in tumorigenesis [12,13].

Several recent studies demonstrated a link between malignancies such as oral squamous cell carcinoma (OSCC) and the oral microbiome [47,48,49]. Interestingly, such studies have also linked the effects of alcohol consumption on the oral microbiome and tumorigenesis [50]. The oral microbiota can be quite complex, as it encompasses several different microenvironments; as such, specific microbial species uniquely dominate the oxygenic oral cavity (*e.g., Streptococci* and *Actinomyces* spp.), while others prefer the subgingival region (below the gum line) where oxygen content is more limited (*e.,g. Bacteroides* spp. and *spirochaetes*).[51,52] Of these common oral microbiota, enrichment of *Streptococcus* is found in the oral cavity, stomach, and intestines of gastric cancer patients, suggesting a role for this microbe in tumorigenesis [53]. *Streptococcus* has further been confirmed to play a role in tumor metastasis and in progression of breast cancer migration to the lung via interactions with endothelial cells and induction of vascular inflammation [54]. Interestingly, use of Iranian propolis (produced by honeybees) has been shown to reduce growth of oral *Streptococcus* while in turn displaying cytotoxic effects specific to cancer cells [55]. Increasingly, evidence is being recognized of, in particular, the interactions that occur between the oral, lung, and gut microbiomes and their impact on immunoregulation, inflammation, and tumorigenesis, although much remains to be understood [49]. The lung microbiome, in combination with the gut microbiome, has been linked to development of tumorigenesis in the lung along with development of lung metastases from various other primary cancers, demonstrating a possible link between the lung microbiome, gut microbiome, and changes in the lung microenvironment that favor malignancy [56,57]. This interaction between the oral and gut microbiomes with the lung and potentially other distal organ microbiomes is impacted by dietary metabolites produced by oral and gut microbiota; these metabolites can further impact the host cells located in these localized and systemic organ systems, including immunomodulation [58,59]. The immunomodulator effects of the microbiomes and the resulting impact on tumorigenesis are clear, showing, for example, the effects of gut microbes such as *H. pylori*, *Campylobacter jejuni*, and chlamydia infections in mediating chronic inflammation, thereby leading to lymphomas [60]. These distal effects can be far reaching, and evidence continues to mount in support of the interactions between the gut microbiome and various other microbiomes such as the breast microbiome and the role of this cross-talk in breast malignancies that are impacted by micronutrients such as queuine [61]. Further links between the gut and oral microbiomes have been found in association with the skin microbiome and malignancies such as melanoma [62], with the esophageal microbiome and esophageal cancers [63], with the uterine and vaginal microbiomes in gynecological cancers [64,65], and with the prostate microbiome and prostate cancer [66,67,68]. These interactions that occur between different microbiomes, dietary factors, and the host differ between different cancers, as highlighted below (Figure 1). Furthermore, those studies that examined diet, microbiome, and tumorigenesis in specific cancers have been summarized in Table 1.

## 4. Oral and Pharyngeal Cancers

High salt intake (e.g., Chinese style salted fish) associated with nitrosamine formation and Epstein–Barr virus has been correlated with nasopharyngeal cancer, which is most common in Asian, Arctic, Middle East, and North African populations [8,88]. In contrast, increased consumption of high-fiber foods (fruits, vegetables), vitamin C, and folate has been associated with decreased risk of oral and pharyngeal cancers, although this evidence is merely suggestive as confounding environmental factors such as altered smoking and alcohol consumption patterns may influence disease outcomes also [8,88]. These correlative studies, which are often plagued by recall or selection biases, are not well supported by prospective studies; however, more recent mechanistic studies have begun to uncover how specific dietary factors found within certain fruits and vegetables influence tumorigenesis [8]. As well, protein breakdown creates a neutral-alkaline environment that can promote periodontal disease, which is a risk factor for oral cancer and oral cancer survival [70,89]. However, studies are limited, and the mechanisms are not well understood [70].

The oral microbiome has been referred to as the ‘oralome,’ an umbrella term encompassing the dynamic interactions between host cells of the oral cavity and microbial communities [90]. The oral microbiome therefore not only regulates interspecies interactions but also mediates the crosstalk between the microbial community and the oral cavity [90]. The balance between a healthy/homeostatic (i.e., eubiotic) and a diseased (i.e., dysbiotic) state depends on the interactions between microbial species within the oral cavity and between the host and the oralome itself. Head/neck cancer and oral cancers have been shown to have a distinct dysbiotic signature, but direct causality between the oralome and oral cancers remains limited [89,90]. The oral microbiome is predominantly composed of a bacterial biome.

Oral cancer is most associated with and thereby influenced by the oral and gut microbiota [69]. Induction of chronic inflammation by bacterial stimulation is one mechanism, which has been thought to influence pathogenesis via production of inflammatory mediators, causing mutagenesis and uncontrolled cell proliferation [91]. The latter process has been thought to be regulated through activation of the nuclear factor κB (NF-κB) signaling pathway and inhibition of apoptotic pathways [92]. The highest mortality oral cancer remains OSCC which is significantly influenced by the oral microbiota through carcinogenetic modulation of cell metabolism (i.e., regulating changing concentrations of nutrients and vitamins) [69]. This modulation can then promote cytokine production associated with different pathological conditions [69]. *Porphyromonas gingivalis*, *Fusobacterium nucleatum*, and *Prevotella intermedia* are the microbes most significantly associated with OSCC; *P. gingivalis* and *F. nucleatum* have been shown to promote tumor progression in mice, and in OSCC these bacteria increase toll-like receptor 2 (TLR2) and pro-inflammatory cytokines IL-6 and IL-8 production, potentially contributing to disease progression [89]. As well, *P. gingivalis* has been found to increase oral cancer cell invasion and proliferation, increasing myeloid-derived suppressor cells and chemokines (CCL2 and C-X-C motif) [89]. Interestingly, the abundance of *F. periodonticum*, *Parvimonas micra*, *Streptococcus constellatus*, *Haemophilus influenza*, and *Filifactor alocis* increases from stages one to four of OSCC [89]. Additionally, studies have shown that the *Fusobacterium, Peptostrepococcus*, and *Prevotella* genera increase in the periodontal tissues in patients with gingival squamous cell carcinoma [89].

The oral microbiome is an ideal biomarker for oral tumors, compared to other biomarkers, highlighting its possible role as an important immunotherapy agent [47]. Commensal bacteria have been shown to enhance the efficacy of immunotherapy with checkpoint inhibitors where tumor growth can be controlled through combined oral administration of *Bifidobacterium* and programmed cell death protein 1 ligand 1 (PD-L1)-specific antibody therapy [47]. Dietary changes over time, including introduction of dairy products, refined carbohydrates, vegetable oils, and alcohol, have been associated with, but not all causatively linked to, a decline in overall oral health and cancers [70]. Dietary factors have a significant influence on the gut microbiome, and this influence branches out to the oral microbiome, highlighting the critical role of the crosstalk between the human microbiome(s), diet, and disease [70].

## 5. Esophageal Cancers

Esophageal cancers are common in areas of Africa, China, and Iran [93]. Green tea is thought to be anti-tumorigenic; however, scalding hot drinks above 65 °C in temperature, such as coffee and tea, have been associated with increasing risk of esophageal cancers, potentially due to thermal injury [94,95,96,97]. Squamous cell carcinoma is the more common form of esophageal cancer globally, and both smoking and alcohol consumption have been noted as risk factors, with limited evidence supporting the direct effects of micronutrients or dietary factors [8,71,73]. However, some suggest that polycyclic aromatic hydrocarbons from high-temperature foods might increase risk and dietary folate, which can be produced by microbes, and might reduce risk of esophageal squamous cell carcinoma [71,72].

Damage of the esophagus caused by acid reflux (via diet or gastroesophageal reflux disease) complicating Barrett’s esophagus (precursor to esophageal adenocarcinoma) can increase risk of esophageal adenocarcinomas [73,98,99]. The esophageal microbiome in esophageal adenocarcinoma is dominated by lactic acid producing *Lactobacillus* which can acidify the esophageal microenvironment which can be further exacerbated by the hydrogen peroxide produced by these microbes [73]. Furthermore, the functions of the esophageal microbiota are altered in esophageal adenocarcinoma, including an upregulation of cell replication and metabolism along with a decrease in fatty acid biosynthesis and D-alanine and nitrogen pathways [73]. Interestingly, while *H. pylori* is best known for its significant role in stomach cancers, there is also an increase in the abundance of this pathogen in esophageal tumor tissues [73]. Meanwhile human papillomavirus and Epstein–Barr virus are reported to increase the risk of developing esophageal squamous cell carcinoma [100]. Furthermore, increased *Porphyromonas gingivalis*, and *Fusobacterium*, along with reduced *Streptococcus*, have been identified in esophageal tumor tissues [73]. Both the oral and intestinal microbiota have also been linked to esophageal cancers; increased *Neisseria* and *Streptococcus pneumoniae* were uncovered in esophageal adenocarcinoma, while increased *P. gingivalis*, *Actinomyces*, and *Atopobium* were indicative of high risk of esophageal squamous cell carcinoma [101,102,103]. Alterations of the microbiome in models of esophageal cancers and precancerous lesions have been associated with TLR and NLR inflammatory pathways, demonstrating these pro-inflammatory pathways along with pro-inflammatory cytokines are increased in esophageal malignancy, possibly via interactions with the resident microbiota [73,103].

Interestingly, a high-fat diet in a mouse model of Barrett’s esophagus induced tumors faster than the control diet, likely related to altered microbiota and an increase in neutrophils and cytokines in the esophagus [74]. Importantly, body size had no effect on tumor growth, suggesting diet rather than obesity influences cancer risk [74]. Associations have been identified between diets that are high in red and processed meats and low in fruits, vegetables (leafy greens especially), and cruciferous vegetables with an increased risk of esophageal cancer [72]. As well, N-nitroso from processed foods is associated with increased risk, influencing cell cycle progression (increased cyclinE 1 and cyclinD 1) and epidermal growth factors (increased transform growth factor α and epidermal growth factor receptor) [71]. Further inverse associations identified include dietary fiber, vitamin E, vitamin C, and β-carotene intake and esophageal cancer [72]. Studies have shown that fiber intake can change the composition of the esophageal microbiome, increasing Firmicutes and decreasing gram-negative bacteria, with limited detection of SCFA-producing bacteria regardless of fiber intake [75]. Disease progression specifically has been worsened by sugar, which increases pro-inflammatory cytokines and causes dysbiosis [71,75].

## 6. Stomach Cancers

Stomach cancers are most commonly diagnosed in Eastern Asia and comprise the fifth most common cancer worldwide [93]. Similar to oral cancers, a diet high in salted foods (e.g., salt preserved fish) may increase risk directly or via nitrates, nitrites, and N-nitroso compounds commonly contained in preserved high-salt foods, smoked foods, and as food additives [76,104,105]. As well, these nitrates and nitrites can mix with heme irons, amines, and amides from other foods to produce N-nitroso compounds [76]. For example, consuming large amounts of pickled foods is thought to increase stomach cancer risk via fungal species commonly found in these foods that produce N-nitroso compounds and also inhibit prostaglandin E synthesis, which protects the mucosa [76,106]. As well, a strong correlation has been identified between increased alcohol intake and risk of gastric cancer [76]. The metabolism of alcohol produces reactive oxygen species (ROS), which can block blood vessels and promote inflammation and injury, and acetaldehyde, which binds to and inhibits DNA synthesis in gastric glands [76]. Again, some correlative evidence supports the benefits of a diet high in fruits, vegetables, and vitamin C in reducing cancer risk, although mechanisms have yet to be uncovered [107]. However, it has been suggested that vitamin C can act as an enzyme cofactor and ROS scavenger, decreasing oxidative damage [76]. Studies show mixed results about the effects of vitamin E on gastric cancer risk; however, one study demonstrated that vitamin E succinate (a vitamin E derivative and by-product of fiber fermentation by microbiota) could induce autophagy in gastric cancer cell lines [76]. Interestingly, supplementation with combinations of α tocopherol, β carotene, vitamin C, and selenium in clinical studies resulted in a regression of precancerous lesions and significant reduction in stomach cancer mortality [108,109,110]. Further, it has been suggested that carotenoids drive a shift from a Th-1 response to a Th-1/2 response, reducing inflammation [76]. As well, green tea consumption in non-smoking women has been shown to reduce risk of stomach cancer, potentially linked to the polyphenols in tea [111]. *Helicobacter pylori* infection directly causes stomach cancer by inducing chronic inflammation and causing DNA damage by converting nitrogen compounds into N-nitroso in gastric fluids, explaining why salted foods might promote tumorigenesis [77,78,112]. Interestingly, studies of the gastric microbiome have been limited due to difficulty culturing most microorganisms residing in the stomach; however, microbial diversity is thought to be significantly lower in patients with gastric cancer [105]. In particular, studies have highlighted an increase in acid-producing microbiota genera including *Lactobacillus* and *Lactococcus*, along with higher pro-inflammatory and pathobiont microbes such as *Fusobacterium*, *Veillonella*, *Leptotrichia*, *Haemophilus*, and *Campylobacter* in gastric cancer patients [105]. Recent advances in sequencing technologies have allowed for the detection of these microbes that make up the gastric microbiome, which have helped improve our understanding of microbes involved in gastric cancers [105].

## 7. Intestinal Cancers

The gut microbiome is the most well studied of the microbiomes to date, demonstrating direct links to development of colorectal cancer which is the third most common cancer globally [93,113]. A wealth of evidence further supports obesity, alcohol, and smoking as risk factors of colorectal cancer, while consumption of processed meat and unprocessed red meat have been significantly linked to carcinogenesis [8,114,115,116,117,118,119,120]. Use of nitrates and nitrites in meat preservatives may expose the gut to mutagenic N-nitroso compounds, while heme iron contained in red meats may also increase N-nitroso and resulting cytotoxicity and gut damage [8]. Interestingly, exposure of certain meats to high heat during cooking can increase mutagenic heterocyclic amines and polycyclic aromatic hydrocarbons, while lactic acid bacteria (primarily *Lactobacillus helveticus* and *Streptococcus thermophilus* and less so *Lactobacillus kefir* and *Lactobacillus plantarum*) in the gut are capable of binding to these chemicals and reducing their mutagenic potential [8,118,121]. Calcium, including in milk products which are associated with moderate reduction in colorectal cancer, may also bind secondary bile acids and heme, reducing their tumorigenic potential [122,123,124].

Consumption of greater than 10 g of total dietary fibers (found in fruits, vegetables, and grains) a day is associated with reduced risk of colorectal cancer [125,126]. Interestingly, humans do not digest dietary fibers; they require gut microbes to ferment them into by-products such as SCFAs, which display beneficial and anti-inflammatory effects in the gut and systemically reduce risk of colorectal cancer [127,128,129]. However, recent evidence suggests there is a pro-inflammatory impact of specific dietary fibers in settings where the gut microbiota is altered, suggesting the microbiota are key to mediating these diet-associated benefits [79]. The opportunities to utilize microbe-altering therapies to manipulate production of SCFA levels in the intestinal tract for the treatment and prevention of cancers, including colorectal cancer could be considered [129,130,131]. This also highlights the implications of dysbiosis (altered microbiota composition) in development and progression of colorectal cancer, as commensal microbes that are key for fiber fermentation (*Bifidobacterium*, *Faecalibacterium*, and *Blautia*) are commonly reduced in colorectal cancer patients [80,81]. Significant associative changes between the gut microbiome and host factors have been highlighted in a recent study of healthy controls, irritable bowel syndrome, inflammatory bowel disease, and colorectal cancer patients, demonstrating increased *Parvimonas*, *Bacteroides fragilis*, Peptostreptococcaceae, and *Streptococcus* spp. associated with Syndecan-1, DNA replication, and cell cycle pathways [132]. Significantly greater details of the highlighted similarities and differences in associations between microbes and host genes in these GI disorders are discussed in the referenced manuscript [132].

## 8. Liver Cancers

Several factors have been directly linked to development and progression of liver cancers, particularly consumption of alcohol which induces liver inflammation associated with cirrhosis (including systemic inflammation) and alcoholic hepatitis [8,133]. Furthermore, alcohol can alter the epithelial barrier of the gut allowing microbes and microbial metabolites (including toxins) to be taken up more readily and translocate to the liver where they are capable of inducing inflammation and subsequently fibrosis and cirrhosis [134,135].

Another disease called non-alcoholic fatty liver disease (NAFLD) has the potential to progress into non-alcoholic steatohepatitis (NASH), which causes inflammation and damage due to excess fat stored by liver cells, cirrhosis, or hepatocellular carcinoma (HCC) [136]. Recent studies have shown data about the role of the gut microbiome in the etiology of NAFLD [82,136,137]. NAFLD is caused by the accumulation of triglycerides (TG) hepatocytes formed from the esterification of fatty acids in the liver [138]. During gut dysbiosis, gut permeability increases, allowing the increased absorption of fatty acids and translocation of bacteria and inflammatory cytokine and leading to worsened inflammation. SCFA producers and fiber fermenters such as *Faecalibacterium prausnitzii* and *Akkermansia muciniphila* are reduced in NAFLD [82]. This lowered production of SCFA could in turn increase gut permeability. Patients with NAFLD also have increased population of *Escherichia coli* that in turn leads to an increased production of ethanol in anaerobic conditions, stimulating the NF-kB pathway that leads to inflammation [82,83].

Furthermore, the fungal species *Aspergillus*, which is commonly found in foods such as dried fruits, nuts, and grains when stored incorrectly (hot and humid), produces a mutagenic toxin, aflatoxin [8]. This is particularly considered a risk factor for individuals with active HBV and HCV infections, which are the primary microbial cause of liver cancers [8]. Meanwhile, some studies suggest consumption of coffee with the bioactive compounds found in coffee beans may reduce risk of liver cancers [8,139,140,141]. Consumption of caffeine increases *Bifidobacterium* species as well as the expression of Aquaporin 8 in the colon, which reduces the risk of cirrhosis and HCC as well as improves barrier integrity [84]. Similarly, green tea extracts have been shown to lead to improved liver enzymes, to reduced body fat, and to an increase in barrier function [142]. The catechins in green tea extract are poorly absorbed and therefore degraded by gut microbes such as *Bifidobacterium*, *Lactobacillus*, and *Ruminococcus*, producing SCFA [143].

On the other hand, a study found that certain fermentable fibers such as inulin induced cholestasis and HCC in dysbiotic mouse models [85]. After receiving a diet rich in soluble fermentable fibers, the HB mice had lower abundance of *Tenericutes* and increased abundance of Proteobacteria, which have been indicated in hepatocarcinogenesis in humans [85]. In addition, there was an increase in *Clostridia*, which are a fiber-fermenting species that also increased the amount of butyrate and secondary bile acids that further aggravated the disease and created a tumor-promoting environment when in a large amount [85].

## 9. Pancreatic Cancers

Pancreatic cancer remains one of the most lethal cancers globally, with surgery as the only potentially curative option for intervention [144]. Several factors associated with obesity increase risk of pancreatic cancer, including diabetes (heightened insulin), suggesting a role for diet [8,145]. Recent research has begun to uncover these links as high-fat diets significantly increase pancreatic metastases and activate receptors involved in driving progression of precancerous pancreatic lesions to pancreatic cancer [146,147,148]. A high-fat diet readily leads to obesity and elicits changes in the gut microbiome and microbial metabolites [149]. In addition, it might lead to the translocation of intestinal microbes as well as detrimental metabolites into the blood stream, which then make their way to the pancreas [150]. This dysbiosis has been associated with tumorigenesis and more aggressive pancreatic cancer [144]. For instance, *Helicobacter pylori*, which has been associated with increased risk of pancreatic cancer, can bind to epithelial cells in the stomach using the adhesin HopQ and carcinoembryonic antigen-related cell adhesion molecules (CEACAM) and inject its virulence factor CagA into the epithelial cells. This then activates a signaling pathway called the Wnt/β-catenin pathway, which is involved in various cellular functions such as proliferation [151].

Moreover, although inconsistent, some studies have also associated the increased intake of red meat and processed meat containing carcinogenic nitrites and N-nitroso compounds (NOCs) with the risk of developing pancreatic cancer due to their ability to form DNA adducts, inducing mutations [152,153]. On the other hand, increased fruit and vegetable intake have been shown to reduce the risk of pancreatic cancer [154,155]. Black raspberries, for instance, have been found to inhibit inflammation, cell transformation, and tumor-specific gene expression as well as increase tumor-infiltrating CD8+ T cells in pancreatic ductal adenocarcinoma [156]. Similarly, another study saw a down-regulation in the miRNA gene responsible for the development of inflammation, metabolic disease, carcinoma, invasion, and metastasis after introducing resistant starch diet in xenograft mice models [157]. This suggests that microbe-altering therapies such as prebiotics, probiotics (e.g., *Faecalibacterium prausnitzii* and *Lactobacillus casei*), and fecal microbiota transplant could offer potential to improve pancreatic cancer outcomes by reducing severity and improving treatment response [144,150,158].

## 10. Breast and Prostate Cancers

The second most common cancer globally is breast cancer, while prostate cancer is the fourth most common global cancer [8,93]. Interestingly, while hormonal factors including estrogen, testosterone, and progesterone are key determinants of risk, the intestinal microbiome has been identified as a major regulator of circulating estrogen, as a producer of testosterone, and as a source for increased risk of breast and prostate cancer [159,160,161,162,163]. Obesity has been linked to breast cancer risk, likely through increased circulating estrogens which are produced in adipose tissues, and to the aggressiveness of prostate cancer, although the evidence remains controversial, and prospective observational studies have been null [8,164,165,166,167]. Adipose tissue in obese individuals may also secrete high levels of plasminogen activator inhibitor-1 (PAI-1), which inhibits enzymes involved in remodeling tissue and degrading blood clots and has been associated with increased risk of breast cancer [168]. Interestingly, the microbes of the gut are able to translocate to the skin and in turn to the breast tissues altering the breast microbiome which has been shown to impact breast malignancies [61]. As mentioned earlier, the micronutrition queuine, which is produced by microbes, is increased in breast cancers, and modifications of queuine can impact tight junction pathways leading to increased migration, invasion, and metastases of breast cancer [61]. Major changes in the breast microbiome in breast cancers include decreased *Anaerococcus*, *Caulobacter*, *Streptococcus*, *Propionibacterium*, and *Staphylococcus*; these changes were associated with increased oncogenic immune potential [169]. Similarly, in prostate cancer, various dietary factors which have been profoundly linked to prostate malignancies can impact the gut, urinary, and prostate microbiomes, demonstrating a decrease in microbes and microbial metabolites that play a significant role in regulating anti-cancer immune surveillance [67,68].

Dietary studies suggest a benefit of vegetable intake, dietary fibers, and soya isoflavins in relation to breast and prostate cancer, although the evidence remains largely inconclusive [170,171,172,173,174]. These foods increase the production of SCFAs by beneficial bacteria such as *Bifodobacterium* and *F. prausnitzii*, which modulate anti-inflammatory and anti-cancer responses [86,175]. As well, total meat and processed meat intake have been associated with an increased risk of breast and prostate cancer [176]. This could be caused by the fat in meat increasing estrogen production and the meat components (heme iron, heterocyclic amines, polycyclic aromatic hydrocarbons, and N-nitroso compounds) causing DNA damage [176]. Further correlative evidence suggested beneficial effects of tomato lycopene, β carotene, vitamin D, vitamin E, and selenium in prostate cancer; however, the data remain inconclusive, requiring further investigations [8,177,178,179,180].

Although difficult to define, evidence shows that the Mediterranean diet reduces the incidence and mortality of prostate cancer, though the mechanisms have yet to be determined [87]. This diet includes legumes, nuts, vegetables and fruits, fish, and eggs and has been shown to lead to an increase in the abundance of *Lactobacillus* compared to the Western diet [181]. *Lactobacillus* has shown anti-tumor activity in synergistic breast cancer models and increased migration and activation of immune cells in other body sites including the mammary glands [182,183]. Studies also suggest that natural folic acid might be protective of prostate cancer [67]. For example, people with prostate cancer show a fecal microbiome with reduced folic-acid-producing microflora, and folic acid is needed for DNA methylation [67]. Further, diets low in fat, paired with exercise, can change hormone levels and induce apoptosis [87]. For instance, mice on a low-fat diet had reduced prostate-specific serum antigen, inulin, and Igf1 mRNA and showed delayed tumor growth overall [87]. As well, eicosapentaenoic acid and docosahexaenoic acid (e.g., n-3 fatty acids in fish oil) have been shown to reduce the proliferation and invasion of prostate cancer cells [87]. Moreover, polyphenols, especially catechin and isoflavone, have a positive effect on prostate cancer [87]. For example, catechin epigallocatechin-3-gallate found in green tea causes cell arrest and induces apoptosis, and soy isoflavone has a similar structure to 17 β-estradiol, allowing it to bind to the estrogen receptor [87]. As well, microbes can metabolize polyphenols into urolithin A and 5-(3′,4′,5′-trihydroxyphenyl)-γ-valerolactose, which decreases proliferation [87].

Total dairy product intake (especially whole milk) and calcium have been associated with an increased risk of prostate cancer [184]. However, it remains unclear if it is the fat component or non-fat components driving this association [184]. For example, dairy might increase circulating hormones, and the casein protein in milk could increase proliferation [184].

## 11. Tumor Microbiomes

The first comprehensive characterization of the tumor microbiome was recently published in *Science*; therefore, there is not yet any evidence linking the tumor microbiome and diet in tumorigenesis [185]. However, this study compared and contrasted tumor biospecimens with adjacent normal tissues in 1,526 clinical samples, uncovering the distinct organ-specific bacteria associated with breast, lung, ovary, pancreas, melanoma, bone, and brain tumors [185]. Uniquely, these intratumor bacterial species were primarily found in the intracellular compartments of cancer and immune cells [185]. The role of the broader microbiome (bacteria, fungi, viruses) in human cancers was further reviewed, presenting the limited causal role of specific microbes as highlighted throughout the present review [186]. Examining the cancer mycobiome, Narunsky-Haziza et al. examined 17,401 clinical samples (blood, plasma, and tissues) in 35 different types of cancer [187]. The findings were similar, demonstrating intratumor fungi are spatially associated with cancer cells and macrophages, and the tumor mycobiome co-resides with tumor bacteriomes [187]. Recent evidence expanded on this topic, taking a step away from the bulk-tissue approaches implemented by previous studies and instead utilizing in situ spatial-profiling technologies and single-cell RNA sequencing to show that these tumor bacteria reside in highly organized micro-niches that display higher immune-suppression and lower vasculature associated with malignant cells compared to tumor micro-niches devoid of bacteria [188]. These intracellular tumor-resident bacteria have also been shown to promote metastasis, carried by circulating tumor cells to secondary sites [189]. However, the question remains of how diet then impacts these tumor resident microbes if they typically reside in low vascular micro-niches? Is nutrient value provided by the tumor cells and neighboring stromal cells [190,191]? Furthermore, can dietary factors impact these microbes during tumor metastasis to secondary sites (e.g., SCFA in the vasculature)?

## 12. The Take Home Message

The described interactions are complex and indicate similarities along with clear differences between dietary factors, microbiomes, and host responses involved in tumorigenesis in different common cancers, summarized in this review (Figure 2). Host microbiomes have co-evolved with the host immune system, demonstrating the significance of microbiomes as a critical influencing factor on health and disease [46,151]. External factors, such as diet, influence the effect of different microbiomes on host immunity, contributing to homeostasis and overall immune function [46]. Therefore, changes in host microbiome communities can impact the ability for crosstalk between different host microbiomes and immune system to occur, contributing to the pathogenesis of several diseases, such as cancer [39].

Cancer is a multifactorial disease and considered to be the second leading cause of death globally [192]. Microorganisms play critical roles in both protecting against or promoting tumorigenesis [192]. Changes in diet and certain dietary factors can introduce perturbations in the host microbiome by affecting the diversity and regulation of the microbiota. Variable microbiota diversity can result in dietary factors interacting with the immune system differently [28]. An in-depth knowledge of the functions and architecture of various microbiomes, along with their interaction with the host, can support development of individualized dietary guidelines and microbe-altering approaches in future. Furthermore, microbe-based therapies can be used to target specific tumor microenvironments as each microenvironment has its own microbiome community, regulating its intra-tumoral processes [39]. However, uncovering the underlying causative and mechanistic interactions between microbiomes, dietary factors, and tumorigenesis requires further studies utilizing improved methods and study designs. From a microbiome standpoint, Niño et al. demonstrated substantial progression using spatial distribution approaches to uncover localized effects of microbes within the tumor microenvironment [188]. However, diet was not taken into account in these studies. As such, one consideration is to ensure appropriate 24 hr recall diet diaries are collected by patients when microbial samples are obtained whenever possible in clinical cohorts [193]. However, these diet data are simply correlative, and it is essential to follow-up this work with dietary intervention studies [194]. The best dietary intervention study design involves a cross-over event where every participant acts as their own control, consuming both the trial diet and the control diet with a washout period between treatments [194]. Ultimately these studies are well supported by continued mechanistic experiments examining the interactions of both individual dietary factors, microbes, and host populations, along with more physiologically relevant whole diets with microbiota and host communities. As described in this review, while research continues to uncover interesting mechanistic and correlative relationships between diet, microbiomes, and tumorigenesis, the majority of studies only examine two of the three factors at a time, and the majority of the studies completed to date have focused specifically on the gut microbiome. This highlights important opportunities for future studies to examine organ-specific microbiomes and how these communities interact with dietary factors and host cell populations. Elucidating mechanisms or therapies to control and influence these processes can help us up-regulate protective mechanisms and down-regulate harmful mechanisms, providing relief to the host. 

## Figures and Tables

**Figure 1 cancers-15-00521-f001:**
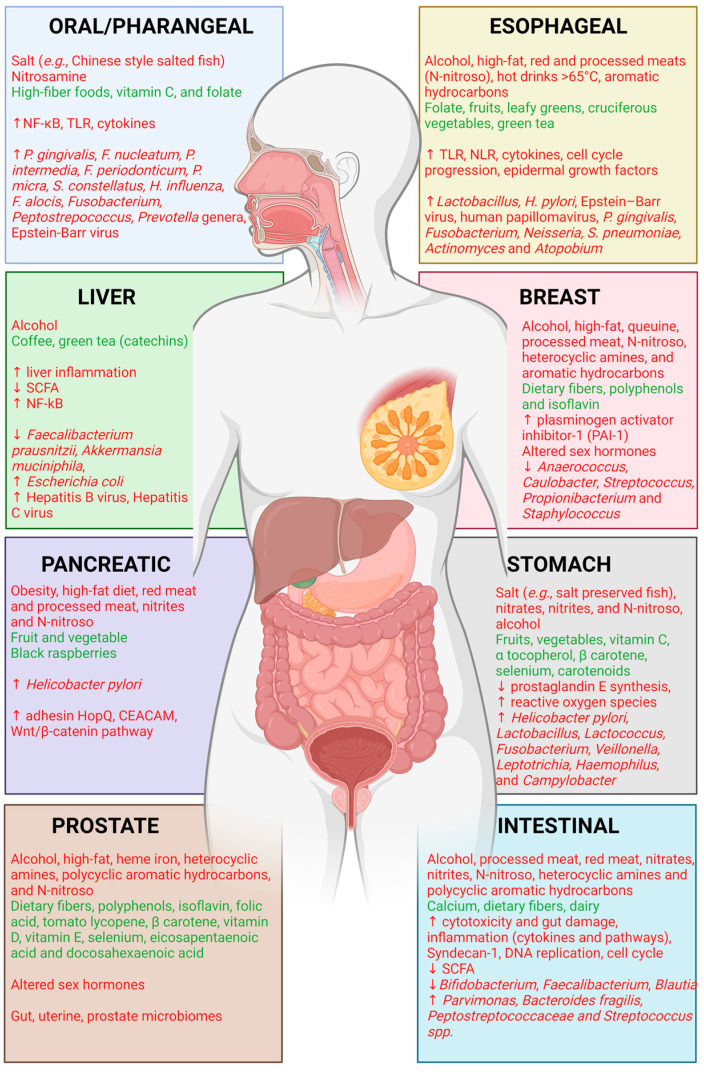
Brief overview of dietary factors, host pathways, and various microbiomes that have been associated with different organ system cancers described and referenced in this manuscript. Red text has been correlated with negative impacts on cancer; green text has been correlated with positive impacts on cancer.

**Figure 2 cancers-15-00521-f002:**
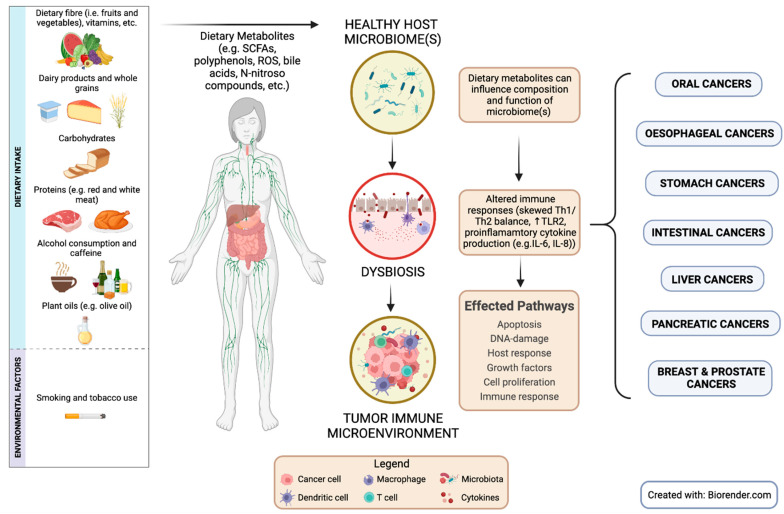
Host microbiomes influence different cancer pathologies via their effects on dietary factors and subsequent immunomodulatory pathways.

**Table 1 cancers-15-00521-t001:** Studies that examined diet, microbiome, and tumorigenesis in specific cancers.

Cancer	Diet Findings	Microbial Findings	Reference
Nasopharyngeal	High salt intake (nitrosamine)	Epstein-Barr virus	[8]
OSCC	Metabolism (nutrients/vitamins)	Oral microbiome	[69]
Oral and pharyngeal	Dietary factors	Gut/oral microbiomes	[70]
Squamous cell carcinoma	Folate	Folate-producing microbes	[71,72]
Esophageal adenocarcinoma	Fatty acid biosynthesis and D-alanine and nitrogen pathways	Esophageal microbiota	[73]
Barrett’s esophagus	High-fat diet	Esophageal microbiota	[74]
Esophageal cancers	Fiber intake, SCFA, sugar	Esophageal microbiota	[75]
Stomach cancers	High salt intake (nitrosamine)	Fungal species	[76]
Stomach cancers	Nitrogens (processed meats)	*Helicobacter pylori*	[77,78]
Inflammatory bowel diseases	Dietary fibers	Fiber-fermenting microbes	[79]
Colorectal cancer	Fiber fermentation	Fiber-fermenting microbes	[80,81]
NAFLD, liver cancer	Fiber fermentation, SCFA	Fiber-fermenting microbes	[82]
NAFLD, liver cancer	Alcohol	*Escherichia coli*	[82,83]
Liver cancers	Dried fruits, nuts and grains	Fungal species (Aspergillus)	[8]
Cirrhosis and HCC	Coffee	Bifidobacterium species	[84]
HCC	Dietary fibers, SCFA (butyrate)	Tenericutes, Proteobacteria, Clostridia	[85]
Breast cancer	Micronutrient queuine	Produced by microbes	[61]
Prostate cancer	Dietary factors	Gut, urinary, and prostate microbiomes	[67,68]
Prostate cancer	Dietary fibers, SCFA	*Bifodobacterium* and *F. prausnitzii*	[86]
Prostate cancer	Folic acid	Folic-acid producing fecal microbes	[67]
Prostate cancer	Polyphenols (catechin/isoflavin)	Gut microbes	[87]

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
