# Peer review of "Host Microbiomes Influence the Effects of Diet on Inflammation and Cancer"

_cancers, 2023, doi:10.3390/cancers15020521_

Round 1

Reviewer 1 Report

This review is a well-written overview of the microbiome-diet-cancer connection. It is very thorough and the authors provide two nice figures to support the review. The review provides an overview of diet and cancer and the microbiome and cancer as it relates to diet. The authors then discuss numerous cancers and the connection to diet and the microbiome. 

1. In the instances of some cancers, studies addressing the microbiome as a mediator between diet and cancer are very sparse. It would be helpful to include a table that outlines the specific studies where this connection has been studied. This table need not include all studies in the review, just those that actually assess the microbiome composition-diet-and cancer.

2. It would also be helpful in the concluding paragraph for the authors to highlight a few specific next steps they feel should be conducted to enhance our ability to answer this question. Some broad applications are addressed, but as there are many gaps in the research for this topic, what do the authors suggest scientists focus on to bridge this gap?

Author Response

Reviewer 1:

This review is a well-written overview of the microbiome-diet-cancer connection. It is very thorough and the authors provide two nice figures to support the review. The review provides an overview of diet and cancer and the microbiome and cancer as it relates to diet. The authors then discuss numerous cancers and the connection to diet and the microbiome.

We thank the reviewer for their feedback and helping us to improve our manuscript to ensure the greatest level of scientific value is expressed.  

  1. In the instances of some cancers, studies addressing the microbiome as a mediator between diet and cancer are very sparse. It would be helpful to include a table that outlines the specific studies where this connection has been studied. This table need not include all studies in the review, just those that actually assess the microbiome composition-diet-and cancer.

We can certainly add a table outlining the specific cited studies that do examine diet and microbiome in the discussed cancers. We have added these references and information into the following table which is now referenced in the manuscript (Table 1).

  1. It would also be helpful in the concluding paragraph for the authors to highlight a few specific next steps they feel should be conducted to enhance our ability to answer this question. Some broad applications are addressed, but as there are many gaps in the research for this topic, what do the authors suggest scientists focus on to bridge this gap?

We thank the reviewer for this feedback and have added a section to the concluding paragraph better highlight some examples of improved methods for studies moving forward.

“However, uncovering the underlying causative and mechanistic interactions between microbiomes, dietary factors, and tumorigenesis requires further studies utilizing improved methods and study designs. From a microbiome standpoint, Niño et al. demonstrated substantial progression using spatial distribution approaches to uncover localized effects of microbes within the tumor microenvironment [190]. Although, diet was not taken into account in these studies. As such, one consideration is to ensure appropriate 24-hr recall diet diaries are collected by patients when microbial samples are obtained whenever possible in clinical cohorts [195]. However, these diet data are simply correlative and it is essential to follow-up this work with dietary intervention studies [196]. The best dietary intervention study design involves a cross-over event where every participant acts as their own control, consuming both the trial diet and the control diet with a washout period between treatments [196]. Ultimately these studies are well supported by continued mechanistic experiments examining the interactions of both individual dietary factors, microbes, and host populations, along with more physiologically relevant whole-diets with microbiota and host communities. As described in this review, while research continues to uncover interesting mechanistic and correlative relationships between diet, microbiomes, and tumorigenesis, the majority of studies only examine 2 of the 3 factors at a time, and majority of the studies completed to date have focused specifically on the gut microbiome. This highlights important opportunities for future studies to examine organ specific microbiomes and how these communities interact with dietary factors and host cell populations.”

Reviewer 2 Report

A comprehensive review covering the role of host microbiomes related to diet, inflammation and cancer.  A very interesting and well written article.

A few minor points for the authors to consider:

1. It would be helpful for readers (who won't necessarily know what each diet consists of) to have the fibre content highlighted (eg high, moderate, low fibre) included for all of the different diet types included in the review, considering the influence of diet on microbiota through available carbon sources (ie fibre).

2. The sentence: 'The gut microbiome has been, and continues to be established as a significant regulator of health and disease, however it is not the only gut microbiome involved in tumorigenesis' (pg 3, line 132) doesn't make sense.  Consider swapping the order of 'the' and 'only' mid sentence.

3. Figure 1 shows a nice overview of content reviewed.  It would be great to have a description of the different colours in text (red vs green) for the regions in the figure caption.

4. For added interest and clarity, it may be beneficial to link the dietary components discussed with roles in increasing/decreasing tumorigenesis for each region with the specific diets these are found in.  Hypothetical example - high fat, associated with x tumour development, associated with a western diet or to some degree a ketogenic diet.  Not sure if it's possible to determine which diets are associated with higher alcohol intake etc?  Would be wonderful to have this depth.

4. One additional microbiome to discuss is the tumour microbiome (separate to specific organ microbiomes where tumours may originate).  Recently identified, thought to also be involved in tumorigenesis.  Here is a link to one recent paper https://www.frontiersin.org/articles/10.3389/fimmu.2022.935846/full#:~:text=Commensal%20bacteria%20and%20other%20microorganisms,%2C%20lung%2C%20and%20breast%20cancers. 

Discussion of the tumour microbiome and it's potential role here would add a further dimension to the interaction of multiple microbiomes, inflammation and tumorigenesis.

Overall this is an informative read, and looks at multiple factors leading to tumorigenesis and the interaction of these factors.

Author Response

Reviewer 2:

A comprehensive review covering the role of host microbiomes related to diet, inflammation and cancer.  A very interesting and well written article.

A few minor points for the authors to consider:

  1. It would be helpful for readers (who won't necessarily know what each diet consists of) to have the fibre content highlighted (eg high, moderate, low fibre) included for all of the different diet types included in the review, considering the influence of diet on microbiota through available carbon sources (ie fibre).

We thank the reviewer for their feedback. While this manuscript is not focused specifically on dietary fibers alone, this is certainly one topic of importance to understanding the impacts of labelled diets on microbiota and host. As such we have added distinctions of fiber volumes to the section discussing mediteranian, keto, and paleo diets (lines 65-83):

“….Mediterranean diet (MD) consist of a recommended intake of dietary carbohydrates (including fibers; ~30g/day), high intake of….”

“….ketogenic diet (KD) which consists of very-low carbohydrate intake (including fibers), high fat intake….”

“….The PD also consists of extraordinarily high amounts of fiber intake (~100g/day), yet interestingly, while the benefits of PD….”

  1. The sentence: 'The gut microbiome has been, and continues to be established as a significant regulator of health and disease, however it is not the only gut microbiome involved in tumorigenesis' (pg 3, line 132) doesn't make sense. Consider swapping the order of 'the' and 'only' mid sentence.

Thank you for noticing this. We believe this was meant to say “it is not the only microbiome”. We have changed this sentence as such.

  1. Figure 1 shows a nice overview of content reviewed. It would be great to have a description of the different colours in text (red vs green) for the regions in the figure caption.

This is an excellent recommendation and we have added this text to the figure legend:

“Figure 1. Brief overview of dietary factors, host pathways and various microbiomes that have been associated with different organ system cancers described and referenced in this manuscript. Red text has been correlated with negative impacts on cancer; green text has been correlated with positive impacts on cancer.”

  1. For added interest and clarity, it may be beneficial to link the dietary components discussed with roles in increasing/decreasing tumorigenesis for each region with the specific diets these are found in. Hypothetical example - high fat, associated with x tumour development, associated with a western diet or to some degree a ketogenic diet. Not sure if it's possible to determine which diets are associated with higher alcohol intake etc?  Would be wonderful to have this depth.

We thank the reviewer for their feedback and certainly agree it would be interesting to be able to place broader labels on whole-food diets associated with different cancers as suggested, although we do not believe this is truly feasible based on current scientific evidence. We have discussed the evidenced links between specific diets and specific cancers within the manuscript. Unfortunately, while these types of speculations could be made, whole-food diets are much more complex than only one or two food group, and there is not a diet necessarily associated with alcohol for example.

For example, recent evidence that red meat consumption in combination with extra virgin olive oil appears to nulify the negative impacts of red meats, therefore we should not assume that every diet high in red-meat is associated with cancer risk. We believe it would therefore be misleading to assume that high-fat correlating with tumorigenesis equates to any diet that is high in fat leading to tumorigenesis. Instead, it is important to design more appropriate studies examining the clinical impact of specific (strictly followed) diets. We have however, made reference to a manuscript that provides a figure of the similarities and differences between the various whole-food diets for those readers who are not familiar with the composition of these diets to improve clarity as we agree this is certainly important:

“The key components included in Mediterranean, Western, Paleolithic, and ketogenic diets and their impacts on gut microbiota have been described previously which Sinibaldi et al. highlighted in Figure 1 of their manuscript [20].”

  1. One additional microbiome to discuss is the tumour microbiome (separate to specific organ microbiomes where tumours may originate). Recently identified, thought to also be involved in tumorigenesis. Here is a link to one recent paper https://www.frontiersin.org/articles/10.3389/fimmu.2022.935846/full#:~:text=Commensal%20bacteria%20and%20other%20microorganisms,%2C%20lung%2C%20and%20breast%20cancers.

Discussion of the tumour microbiome and it's potential role here would add a further dimension to the interaction of multiple microbiomes, inflammation and tumorigenesis. Overall this is an informative read, and looks at multiple factors leading to tumorigenesis and the interaction of these factors.

We thank the reviewer for this recommendation. There is certainly an entire world of “tumour microbiome” research progressively being explored in journals such as Cell and Nature. While we excluded these articles previously due to the lack of understanding how diet is implicated with these tumor microbiomes in tumorigenesis, we agree that brief discussion of these works can be included here. As such we have added a brief section regarding tumour microbiomes:

11. Tumor microbiomes

The first comprehensive characterization of the tumor microbiome was recently published in Science, therefore there is not yet any evidence linking the tumor microbiome and diet in tumorigenesis [186]. However, this study compared and contrasted tumor biospecimens with adjacent normal tissues in 1,526 clinical samples, uncovering the distinct organ-specific bacteria associated with breast, lung, ovary, pancreas, melanoma, bone, and brain tumors [186]. Uniquely, these intratumor bacterial species were primarily found in the intracellular compartments of cancer and immune cells [186]. The role of the broader microbiome (bacteria, fungi, viruses) in human cancers were further reviewed, presenting the limited causal role of specific microbes, as highlighted throughout the present review [187]. Examining the cancer mycobiome, Narunsky-Haziza et al. examined 17,401 clinical samples (blood, plasma, tissues) in 35 different types of cancer [188]. The findings were similar, demonstrating intratumor fungi are spatially associated with cancer cells and macrophages and the tumor mycobiome co-resides with tumor bacteriomes [188]. Recent evidence expanded on this topic, taking a step away from the bulk-tissue approaches implemented by previous studies, and instead utilizing in situ spatial-profiling technologies and single-cell RNA sequencing to show that these tumor bacteria reside in highly organized micro-niches that display higher immune-suppression and lower vasculature associated with malignant cells compared to tumor micro-niches devoid of bacteria [189]. These intracellular tumor-resident bacteria have also been shown to promote metastasis, carried by circulating tumor cells to secondary sites [190]. Yet, the question remains, how does diet then impact these tumor resident microbes if they typically reside in low vascular micro-niches? Is nutrient value provided by the tumor cells and neighboring stromal cells [191, 192]? Furthermore, can dietary factors impact these microbes during tumor metastasis to secondary sites (e.g., SCFA in the vasculature)?

We also feel it is important to mention for full disclosure, Dr Armstrong very recently reviewed a manuscript for Cancers journal which is under revision and is a review of tumour microbiomes. We do not want to create a conflict of interest situation. As such we have ensured a brief overview of tumor microbiomes has been added and highlight the lack of connection to diet as of yet, as described above. We hope this section will suffice and appreciate the editor and reviewers feedback.
